# Analysis of the glyco-code in pancreatic ductal adenocarcinoma identifies glycan-mediated immune regulatory circuits

Ernesto Rodriguez [1], Kelly Boelaars[1], Kari Brown[1], Katarina Madunić[2], Thomas van Ee[1], Frederike Dijk[3], Joanne Verheij[3], R. J. Eveline Li[1], Sjoerd T. T. Schetters [1], Laura L. Meijer [4], Tessa Y. S. Le Large [4], Else Driehuis[5], Hans Clevers [5], Sven C. M. Bruijns[1], Tom O'Toole[1], Sandra J. van Vliet[1], Maarten F. Bijlsma[6,7], Manfred Wuhrer [2], Geert Kazemier[4], Elisa Giovannetti [8,9], Juan J. Garcia-Vallejo[1] & Yvette van Kooyk [1✉]

Pancreatic ductal adenocarcinoma (PDAC) remains one of the most aggressive malignancies with a 5-year survival rate of only 9%. Despite the fact that changes in glycosylation patterns during tumour progression have been reported, no systematic approach has been conducted to evaluate its potential for patient stratification. By analysing publicly available transcriptomic data of patient samples and cell lines, we identified here two specific glycan profiles in PDAC that correlated with progression, clinical outcome and epithelial to mesenchymal transition (EMT) status. These different glycan profiles, confirmed by glycomics, can be distinguished by the expression of O-glycan fucosylated structures, present only in epithelial cells and regulated by the expression of GALNT3. Moreover, these fucosylated glycans can serve as ligands for DC-SIGN positive tumour-associated macrophages, modulating their activation and inducing the production of IL-10. Our results show mechanisms by which the glyco-code contributes to the tolerogenic microenvironment in PDAC.

[1] Amsterdam UMC, Vrije Universiteit Amsterdam, Department of Molecular Cell Biology and Immunology, Cancer Center Amsterdam, Amsterdam Infection and Immunity Institute, De Boelelaan 1117, Amsterdam, The Netherlands. [2] Center for Proteomics and Metabolomics, Leiden University Medical Center, Albinusdreef 2, Leiden, The Netherlands. [3] Amsterdam UMC, Academic Medical Center Amsterdam, University of Amsterdam, Department of Pathology, Cancer Center Amsterdam, Meibergdreef 9, Amsterdam, The Netherlands. [4] Amsterdam UMC, Vrije Universiteit Amsterdam, Department of Surgery, Cancer Center Amsterdam, De Boelelaan 1117, Amsterdam, The Netherlands. [5] Oncode Institute, Hubrecht Institute, Royal Netherlands Academy of Arts and Sciences and UMC Utrecht, Utrecht, The Netherlands. [6] Amsterdam UMC, University of Amsterdam, LEXOR, Center for Experimental and Molecular Medicine, Cancer Center Amsterdam, Meibergdreef 9, 1105AZ Amsterdam, The Netherlands. [7] Oncode Institute, Meibergdreef 9, 1105AZ Amsterdam, The Netherlands. [8] Amsterdam UMC, Vrije Universiteit Amsterdam, Department of Medical Oncology, Cancer Center Amsterdam, De Boelelaan 1117, Amsterdam, The Netherlands. [9] Cancer Pharmacology Lab, AIRC Start-Up Unit, Fondazione Pisana per la Scienza, Pisa, Italy. ✉email: y.vankooyk@amsterdamumc.nl

Pancreatic ductal adenocarcinoma (PDAC) represents one of the most aggressive malignancies, with a 5-year survival rate of only 9% and predicted to become the second leading cause of cancer-related deaths in the United States[1,2]. This is mainly a reflection of late diagnosis, as many patients present with surgically unresectable tumours, and a lack of other effective therapies[1,3].

A key hallmark of PDAC is a complex tumour microenvironment (TME) dominated by an extracellular matrix with stromal cells, tumour and immune cells intermixed[4]. The molecular characteristics and architecture of the TME induce a heavy immune suppression that contributes to tumour progression. Strong desmoplastic features, formed by activated pancreatic stellate cells, as well as increased deposition of extracellular matrix facilitates tumour growth and metastasis[4,5]. Additionally, the presence of suppressive immune cells, such as myeloid-derived suppressor cells (MDSCs), tumour-associated macrophages (TAM) and dendritic cells (DC) are controlled by the TME to direct immune suppression[6,7].

Changes in glycosylation are a universal characteristic of malignant transformation and some glycan structures are widely used as cancer biomarkers[8,9]. Currently, one of the most used serum markers for PDAC is the CA19-9, which corresponds to sialyl Lewis[A]. Serum levels of CA19-9 is used for patient follow-up and other glycan structures or glycoproteins have been proposed as markers for early detection and patient stratification[8,10]. Although some glycan changes have been reported for PDAC, a comprehensive overview and understanding of cancer-associated glycan alteration is lacking.

Different glycosylation profiles of cancer cells are strongly associated with cancer progression and immune-modulation[8]. Because immune cells express a large variety of glycan-binding receptors (called lectins), such as C-type lectins (DC-SIGN, MR and MGL), Siglecs and galectins, they can sense and respond to changes in the glycan signature of their environment; which often leads to the induction of inhibitory immune processes in those cells[8,9]. However, it is unclear which glycan signatures present in PDAC are able to interact with immune cells and contribute to the tolerogenic TME. Therefore, a better understanding of glycan expression profiles may provide new insights into mechanisms of immune escape.

During the last decade, the development of transcriptomic and genomic methodologies has drastically improved our current understanding of the molecular events that occur during PDAC development and their relationship with therapy[11–13]. Indeed, PDAC subtypes have been identified which are associated with different clinical outcomes (such as survival and response to therapy)[11–14]. Despite these advances, the contribution of tumour glycosylation has been widely overlooked.

Based on the work of our group and others, we recently defined a group of tumour-associated glycan signatures, which we denominated as the tumour **glyco-code**, able to induce immune-modulatory properties in immune cells by triggering specific lectin-glycan interactions: O-glycosylation, fucosylation, sialylation and galectins[8]. In this article, we performed a deep characterisation of the glyco-code in PDAC by analysing publicly available transcriptomic data of patient samples and cell lines and performing glycan profiling using mass spectrometry and lectin staining, providing evidence of the clinical relevance of the PDAC glyco-code and the induction of a tolerogenic microenvironment.

## Results

**The glyco-code in PDAC.** The array of glycan structures expressed by cancer cells depends on a highly regulated network of glycoproteins and enzymes that build up or degrade glycan structures, collectively known as the glycosylation machinery[8,15]. As a starting point of our study to investigate the glycosylation machinery in pancreatic cancer, we designed an analytical pipeline based on transcriptomic analysis of glycosylation-related genes using publicly available datasets, focusing on genes that directly contribute to the pathways associated with the *glyco-code* (O-glycosylation, fucosylation, sialylation and galectins) (Fig. 1a and Table S1). Given the complex nature of the glycosylation machinery and the different levels of regulation that possesses, gene expression is not always able to predict the specific glycan structures exposed on the membrane nor their abundance, but have been shown to identify activated glycosylation pathways in cells[15]. In particular, the transcriptomic analysis allowed us to explore glycosylations alterations in clinical samples and identify potential subgroups in cancer patients based on the expression of genes associated with the *glyco-code*.

We started by performing a differential expression analysis of glyco-code related genes between normal and tumour tissue in four transcriptomic datasets (whose characteristics can be found in Table S2). This analysis revealed a series of differentially expressed glyco-code genes in PDAC compared to normal pancreatic tissue, in particular those involved in the initiation of the O-glycosylation pathway (Fig. 1b). Using this strategy, we previously reported that PDAC present an increased expression of sialylation-related genes, specially the synthesis of the sialic acid donor (CMP-Neu5Ac), as illustrated in Fig. S1a[16]. These had the characteristics of increased expression of apomucins (*MUC1, 4, 5B, 13, 16, 17* and *20*), proteins known for being heavily O-glycosylated, as well as of different polypeptide N-acetylgalactosaminyltransferases (*GALNT3, 5, 6, 10* and *12*), a family of enzymes that catalyse the first step of the O-glycosylation pathway. Interestingly, most of the remaining genes differentially expressed in the tumour could also be integrated into the same pathway (Fig. 1b), predictably leading to: (I) sialylation of truncated O-glycans, including the synthesis of the antigens sialyl-Tn ① or sialyl-T ②, or (II) extension and synthesis of LacNAc structures, which can subsequently be sialylated and/or fucosylated ③. Several of these structures, such as the sialyl-Tn antigen or CA19-9 (sialyl-Lewis A), are known to be upregulated in pancreatic cancer, whereby CA19-9 is used in the clinic as a tumour marker to monitor disease progression[10]. Our analysis also showed an increase in the genes encoding for enzymes involved in the synthesis of the fucose donor (GDP-Fuc, Fig. 1b and S1a), suggesting a general increase of fucosylation in PDAC. Conversely, only mild changes were seen in the expression of N glycosylation genes compared to normal tissue (Fig. S1b).

**The tumour glyco-code defines molecular subtypes in pancreatic cancer.** The molecular characterisation of pancreatic cancer has recently led to the identification of several clinically relevant subtypes[11–14]. To study whether the expression of glyco-code related genes could also characterise different molecular subtypes, we performed consensus clustering in three discovery datasets (ICGC-PACA-AU, TCGA-PAAD and E-MTAB-6134), using the differentially expressed genes found in our previous analysis. This approach allowed us to identify three different glyco-code based clusters characterised by the expression of specific glycan gene signatures (Fig. 2a–c and Fig. S2a, e). Network and gene set enrichment analysis showed high interconnectivity between the glyco-code related clusters and the subtypes previously identified, with particular overlap with the 'Classical' and 'Basal-like' subtypes defined by Moffit and collaborators[12] (Fig. 2d, e and Fig. S2i).

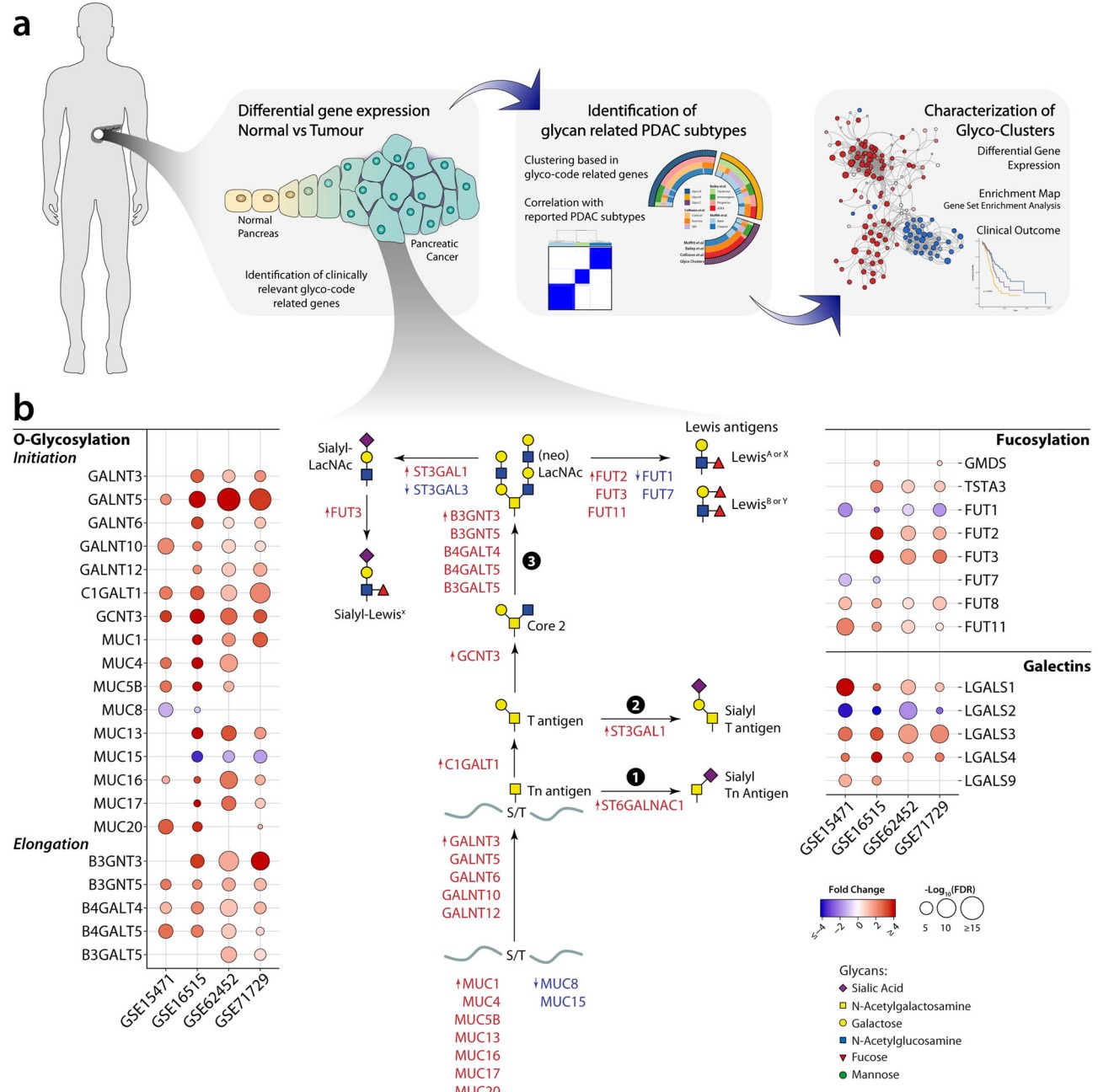

**Fig. 1 Transcriptional changes in glyco-code related genes in pancreatic cancer. a** Diagram of the steps used in this paper for the analysis of the role of the glyco-code in pancreatic cancer. **b** Analysis of glyco-code related genes differentially expressed in tumour tissue with respect to their normal counterpart showed fundamental changes in pancreatic cancer. Bubble plot representing the differentially expressed glyco-code related genes (rows) y four different datasets. Colour of the circles is associated with the fold change of tumour vs normal: red shows genes upregulated in tumour samples, while blue in normal samples. The size of the circles is associated with the $p$ value, calculated as $-\log_{10}(p$ value).

*Cluster A—Fucosylated subtype.* This subtype is characterised by increased expression of genes involved in fucosylation (*GMDS, FUT2, FUT3* and *TSTA3*) and O-glycosylation (*GALNT4, GALNT6, MUC1, MUC13* and *MUC17*; Fig. 2b). Gene set enrichment analysis showed an increased fucosylation pathway with a higher expression of genes encoding mucins and enzymes involved in the extension of O-glycans (Fig. 2c and Fig. S2c, g), suggesting the expression of complex structures. Network analysis shows an association of this cluster with the Progenitor subtype defined by Bailey et al. and the Classical subtypes defined by Collisson et al. and Moffit et al.[11,12] (Fig. 2d and Fig. S2b, f). This correlation was confirmed by gene set enrichment analysis

(Fig. 2e). Survival analysis shows a better prognosis for patients belonging to this cluster compared to other PDAC patients (Fig. 2f).

*Cluster B—Basal subtype.* Patients classified in this subtype present a higher expression of genes encoding galectin-1 (*LGALS1*) and the mucins *MUC4* and *MUC16* (Fig. 2b). This is associated with previously described PDAC subtypes characterised by a poor prognosis, such as the Basal subtype defined by Moffitt et al. (for which is named), but also the quasi-mesenchymal in Collisson et al. and squamous in Bailey et al. (Fig. 2d, e and Fig. S2i). Interestingly, these subtypes have been associated with a high

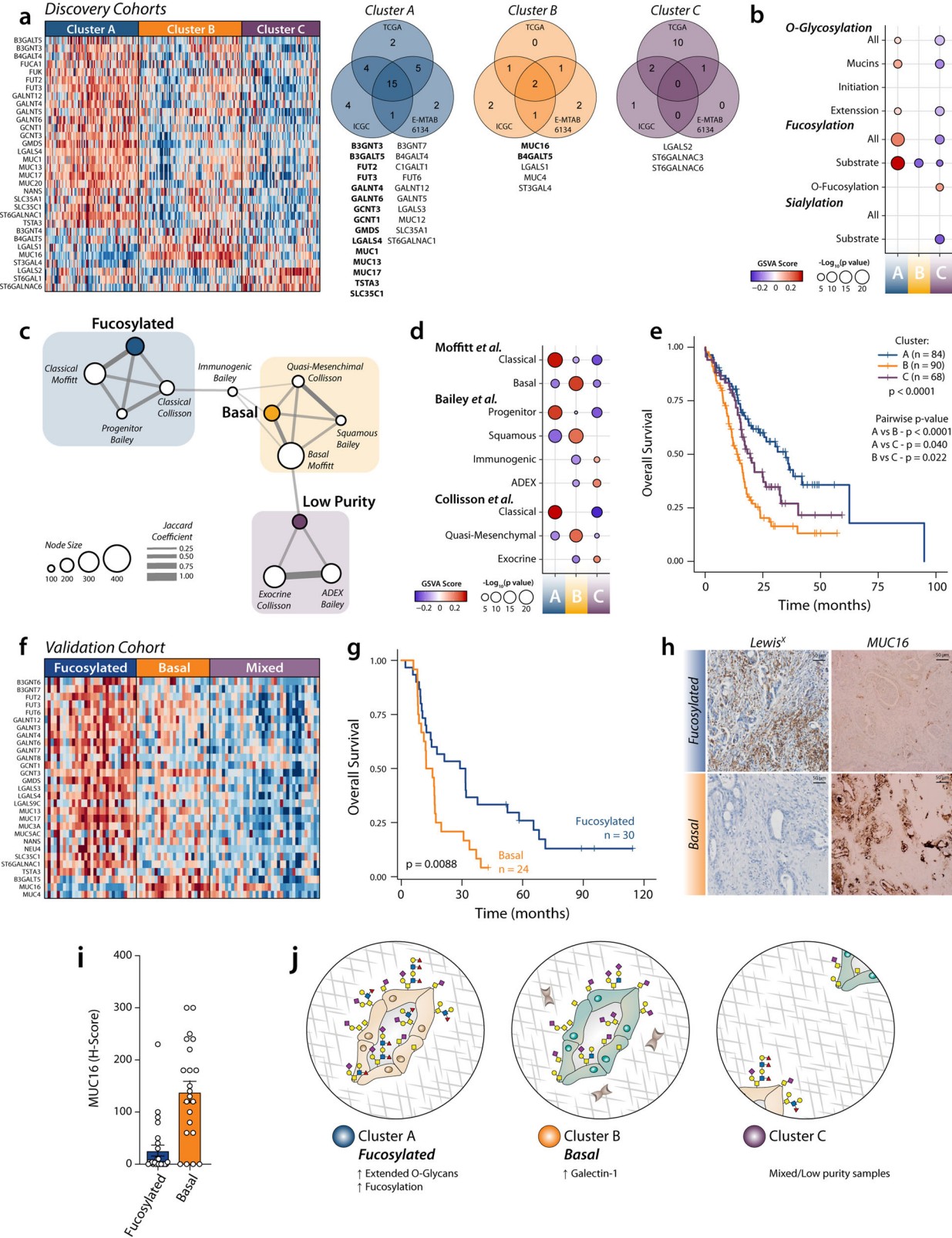

epithelial to mesenchymal transition (EMT)[17]. Keeping with these findings, Galectin-1 and MUC16 have been associated with EMT in different kinds of cancer, including PDAC[18–20]. In addition, the *Basal* subtype was associated with poorly differentiated tumours (Fig. S2j), which has also been previously associated with EMT in PDAC[17,21].

*Cluster C—Mixed/low tumour content.* This cluster is correlated with the subtypes Exocrine (described by Collisson et al.) and ADEX (Bailey et al.) (Fig. 2d). In our analysis, no specific glycan signature was associated with this cluster. Evaluation of tumour cellularity showed that this cluster mainly represented low purity samples (Fig. S2k, l), in line with previous reports that show that

**Fig. 2 Expression of glyco-code related genes define molecular subtypes with differential prognosis. a** Consensus clustering revealed the presence of three different glyco-clusters of patients with pancreatic cancer. Heatmap of differential expressed genes between the different clusters in the ICGC-PAAD-AU dataset. Right: Venn diagram showing the common differential expressed glyco-code related genes in the identified clusters in the discovery datasets. Genes identified in at least two datasets are listed (in bold letters, genes identified in all three datasets). **b** GSVA scores for different glycosylation pathways associated with the glyco-code. **c** Network analysis of interconnectivity between the identified clusters and previously reported molecular subtypes. **d** Enrichment scores using GSVA for different gene sets associated with the subtypes described by Bailey et al., Moffit et al. and Collisson et al. **e** Survival analysis revealed differences in prognosis in the different subtypes. Statistical analysis: Log-rank test. **f** Heatmap of differential expressed genes between the different clusters in the validation cohort. **g** Survival analysis of the fucosylated and basal subtypes. Statistical analysis: Log-rank test. **h** Immunohistochemistry of MUC16 in PDAC tissues from patients in the validation cohort classified as a basal or fucosylated subtype. Data presented as mean values ± SEM. **i** Quantification of the staining of MUC16 in PDAC tissues. **j** Diagram summarising the characteristics of each subtype.

high-purity samples present a binary classification in the Basal and Classical Subtypes defined by Moffit et al.[14]. This data also suggests that the tumour glyco-code subtypes are mainly defined by the presence of the cancer cells.

To further characterise the glyco-clusters, we analysed datasets containing transcriptomic data from normal and tumour tissue, which allowed us to study the differential expression of genes between each subtype and the corresponding normal tissue (Fig. S3). This analysis confirmed the particular increased expression of fucosylation and *O*-glycan extension-related enzymes in the Cluster A, while several genes associated with sialylation and the initiation of *O*-glycosylation were increased in both the *Fucosylated* (Cluster A) and the *Basal* (Cluster B) subtypes (Fig. S3). Interestingly, few changes in the expression of glyco-code-related genes were found between Cluster C and normal tissue (Fig. S3d, i), further supporting the idea that the cancer cells are the main contributors to the glyco-code clusters. Given this, we continue working on the characterisation of the *Fucosylated* and the *Basal* subtypes.

To validate our results, we performed analysis of the RNA-seq data from an independent cohort of PDAC patients with matched tissues recently published[22]. First, we confirmed the presence of *Basal* and *Fucosylated* subtypes in this new dataset, observing differences in overall survival as found in the previous analysis (Fig. 2g, h). We next used immunohistochemistry to confirm the presence of the fucosylated antigen Lewis[X], as a marker of the *Fucosylated* subtype, and MUC16, specific for the *Basal* subtype (Fig. 2i).

In summary, our analysis showed that changes in the glyco-code during PDAC progression mainly define two specific tumour-subtypes that can be differentiated by the expression of complex *O*-glycans and fucosylated structures (Fig. 2j).

**The Basal subtype is characterised by a high EMT status.** Given the fact that our results suggest that the tumour glyco-code is mainly driven by cancer cells, we continued our analysis using transcriptomic data from laser microdissected samples (ICGC-PACA-CA), to focus on molecular characteristics of tumour cells and excluding the stromal contribution. The implementation of consensus clustering and gene enrichment analysis allowed us to confirm the presence of the *Fucosylated* and *Basal* subtypes as we described Fig. 2, that also presents differential clinical outcomes (Fig. S4a–d).

A deeper characterisation of the biology of both subtypes was performed using gene set enrichment analysis (GSEA) and employing Enrichment Map for visualisation, which allows to represent GSEA results in network layout where redundant gene sets are clustered by overlap (Fig. S4e). Interestingly, we found that the gene set most enriched in the *Basal* subtype is the 'HALLMARK Epithelial to Mesenchymal Transition' (Fig. 3a–c). In agreement with our transcriptomic analysis of bulk tissue (Fig. 2), cluster associated with higher EMT phenotype in cancer

cells present poor survival (Fig. S4d). To confirm this association, we defined an EMT Score, based on the difference between a Mesenchymal and an Epithelial Score, which was indeed increased in the *Basal* subtype (Fig. 3c). When focusing on the glyco-code related gene sets, we also saw an enrichment of *O*-glycosylation and fucosylation-related pathways, as expected (Fig. S4f).

EMT can be defined as a reversible continuum between the epithelial and the mesenchymal states, however, also intermediate states can be found. To study how glycosylation changes during EMT, we correlated the EMT Score with the expression of different glyco-code-related genes. As expected, we found a negative correlation between several genes associated with *O*-glycosylation (*MUC13*, *GALNT7*, *GALNT3* and *MUC20*) and fucosylation (*GMDS*, *TSTA3*, *FUT2*, *FUT3* and *FUT6*), suggesting that the latter genes are associated with an epithelial cell phenotype (Fig. S4g, h). On the other hand, Galectin-1 (*LGALS1*) was the glyco-code related gene with a higher positive correlation with the EMT status, consistent with our differential gene expression analysis (Fig. S4g, h).

Organoid technology represents a novel in vitro tool to study cancer biology, allowing to establish three-dimensional cultures that resemble tumour characteristics[23]. To analyse if the glyco-code derived subtypes are also reflected in organoids, we performed an analysis of transcriptomic data from PDAC-derived organoids recently published[24]. Interestingly, we could also identify the presence of the *Fucosylated* and *Basal* subtypes, observing an overlap in the genes associated to each cluster with the ones found in tissue (Fig. S4i).

We next wondered if the different patterns of expression of glyco-code related genes associated with epithelial and mesenchymal phenotype are present at a single-cell level and can coexist within the same tumours. For this, we analysed scRNA-seq data previously reported by Peng et al., which contains data from 24 PDAC and 11 Normal samples, and were able to identify four clusters of tumour cells with different degrees of association with the molecular subtypes described by Moffit et al. (Fig. 3d)[12,25]. Quantification of the different clusters showed that Classical and Basal tumour cells can coexist within the same patient, despite that the most extreme phenotypes are particularly associated to specific patients (Fig. 3e). Interestingly, we observed a differential expression of the EMT markers E-Cadherin and Vimentin in the different clusters, corroborating the idea that the Classical and Basal subtype are associated with Epithelial and Mesenchymal phenotypes, respectively. Indeed, we also found a negative correlation between the Classical Score with an EMT Score (Fig. 3f, g). When analysing the expression of fucosylation-related genes, we can observe their association with the Classical subtype also at a single-cell level (Fig. 3f). Interestingly, when evaluating general pathways using gene sets, we found that *O*-glycosylation and fucosylation positively correlated with the Classical score, in agreement with our previous results (Fig. 3h, i). Moreover, we

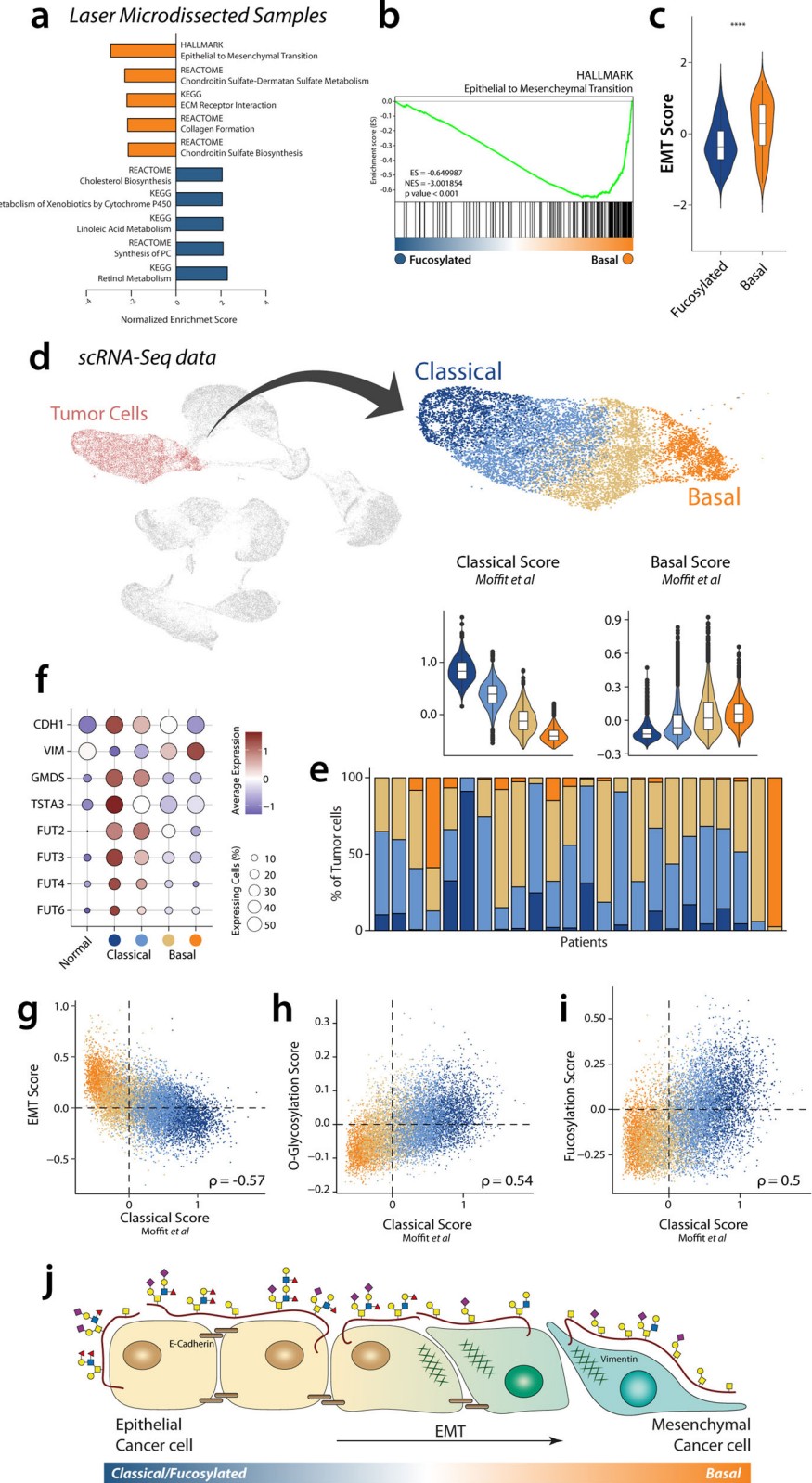

found that genes associated with epithelial phenotype (like *LGALS4*) positively correlated with the Classical Score, while the genes *LGALS1* and *MUC16* are associated with the Basal subtype and present a negative correlation with Classical Score (Fig. 3h and Fig. S4j, k). Overall, our results show that EMT in pancreatic cancer is associated with a decrease in the expression of *O*-glycosylation and fucosylation-related genes (Fig. 3i).

**Glyco-code based subtypes in PDAC cell lines.** Cell lines derived from tumour tissue have served as fundamental tools in cancer research. We next wondered whether the two glyco-code subtypes found in the analysis of PDAC tissue are also reflected in pancreatic cancer cell lines and could be used as tools for functional validation of the glyco-code. Consensus clustering using previously published transcriptomic data allowed us to identify two

**Fig. 3 Glyco-code subtypes are associated with EMT and coexist in PDAC tumours. a** Graphical representation of the top five gene sets enriched in each cluster. **b** GSEA representation of the gene set 'HALLMARK Epithelial to Mesenchymal transition'. **c** EMT Score of both glyco-clusters A and B. Statistics: Mann–Whitney test (*$p \leq 0.05$, **$p \leq 0.01$, ***$p \leq 0.001$, ****$p \leq 0.0001$). **d** Analysis of single-cell RNA-seq identify clusters of tumour cells associated with the Classical and Basal molecular subtypes defined by Moffit et al. Data presented as violin plots with boxplots indicating the median, 25th and 75th percentiles (hinge) and whiskers represent 1.5 times the interquartile range. **e** Quantification of the Classical (blue) and Basal (beige-orange) tumour clusters in different patients. **f** Expression of EMT markers E-Cadherin (CDH1), Vimentin (VIM) and *Fucosylation*-related genes in different clusters in scRNA-seq. Correlation of EMT scores **g** *O*-glycosylation **h** and Fucosylation pathway **i** with Classical score. **j** Our results suggest defined changes in glycosylation during EMT in PDAC.

clusters of PDAC cell lines, indeed associated to the *Basal* and *Fucosylated* subtypes (Fig. S5a–c and Table S3)[26]. This analysis revealed common pathways enriched in the glyco-code subtypes identified in both tumour cell lines and tissue (Fig. S5a, c). Characterisation of these subtypes by differential gene expression and the gene set enrichment analysis confirmed the association of these clusters to different stages in the EMT continuum (Fig. S5a, b).

To confirm that the differential expression of glyco-code-related genes in the different subtypes is indeed reflected in the presence of the predicted glycosylation patterns on the membrane, and thus analysed the presence of different fucose-containing structures in the tumour cell lines using glycan-specific antibodies and specific plant lectins that specifically recognise different fucosylated motives. Interestingly, PDAC cell lines could be grouped based on their recognition by the lectins from *Lotus Tetragonolobus* (LTA) and *Ulex Europaeus* (UEA I) that both recognise fucose-containing structures, in line with our results from the transcriptomic analysis (Fig. 4a). Surprisingly no differences were found in the recognition of the tumour cell lines by other plant lectins with distinct glycan specificity, indicating that the presence or absence of fucose-containing structures is the most distinctive (Fig. S5d). Some of the most interesting fucosylated structures are the Lewis antigens, given their role in the interaction with different lectin receptors (as DC-SIGN and Selectins) and their use in the clinic to monitor disease progression (as described previously for CA19-9)[27]. Indeed, tumour cell lines associated with the *Fucosylated* subtype expressed Lewis antigens, while these structures were barely present in the tumour cell lines with a mesenchymal phenotype (Fig. 4a, b). Other relevant fucosylated antigens, such as CA19-9 and VIM-2, could be found in some of the tumour cell lines within the *Fucosylated* subtype (Fig. S5e–g). Using glycosylation inhibitor, we observed and confirmed that the fucosylated structures found in epithelial tumour cell lines are mostly present in *O*-glycans (which extension is inhibited by benzyl-GalNAc), but not in N-glycans (Inhibited by Kifunensine) (Fig. S5h).

Recently, Huang et al. developed GlycoMaple, a comprehensive glycosylation mapping tool, which allows to visualise different pathways that are active in cells, based on their gene expression[15]. We made use of this tool to visualise the *O*-glycosylation pathway in the different cell lines, which suggests that epithelial cell lines (BxPC3, PaTuS and PL45) present a more diverse array of *O*-glycans than Mesenchymal cells (Mia Paca-2 and PaTuT), consistent with our analysis (Fig. S6).

To confirm this and identify the specific *O*-glycan structures that differentiate the *Basal* and *Fucosylated* subtypes in pancreatic cancer, we performed an *O*-glycomic analysis in the different cell lines (Fig. 4c). For this, *O*-glycans were released via reductive β-elimination and subsequently analysed by porous graphitised carbon (PGC) liquid chromatography-tandem mass spectrometry, as described before[28]. Consistent with our *in silico* analysis, fucosylated *O*-glycans were only observed in epithelial cell lines, which were also characterised by high levels of elongated structures (Fig. 4c, d). On the other hand, the

structures found in *Basal* cell lines were restricted to short *O*-glycans, mainly di-sialylated T antigen. Interestingly, dimensional reduction of the glycan abundance using principal component analysis (PCA) showed two different clusters of *Fucosylated* cell lines, while the *Basal* cell lines cluster together (Fig. S5i). To get a better understanding of the differences in the *Fucosylated* cell lines, we evaluated the glycan structures that contribute to their separation in PCA (Fig. S5j). This revealed that PaTuS present a higher expression of extended Core 2 *O*-glycans than PL45 and BxPC3, which however present mainly Core 1 structures (Fig. 4c and Fig. S5j). This can be explained by the higher expression of *GCNT3* in PaTuS in respect to the other *Fucosylated* cell lines.

By using Western blot and microscopy we verified the expression of EMT associated proteins ZEB1 and vimentin in the *Basal* subtype and the expression of epithelial associated markers (E-Cadherin, RAB25, TACSTD2 and MAL2) in the *Fucosylated* subtype (Fig. 4e, f). As one of the main characteristics of the EMT continuum is its reversibility, we next wondered whether the null expression of fucosylated antigens could be reverted in the *Basal* subtype. For this, we studied the role of the transcription factors ZEB1/2, overexpressed in the *Basal* cell lines and associated to metastasis and cell plasticity in PDAC[21]. Indeed, knockdown of ZEB1 and/or its homologue ZEB2 using siRNA led to an increase in Lewis[x] expression, suggesting that these transcription factors are involved in the suppression of fucosylated glycans once cells undergo EMT (Fig. 4g).

### GALNT3 contributes to the glycosylation phenotype of epithelial cells.

To further characterise the molecular mechanisms involved in the differential expression of fucosylated antigens during EMT, we analysed the glyco-code-related genes differentially expressed in the different subtypes of the cell lines. Interestingly, this analysis revealed that the most differentially expressed glyco-code-related gene in *Lewis*[high] cell lines was the polypeptide N-acetylgalactosaminyltransferase 3 (*GALNT3*), a Golgi-located enzyme that catalyses the addition of GalNAc to serine and threonine in glycoproteins to form the Tn antigen, thereby initiating the *O*-glycosylation pathway (Fig. 5a, b). However, the analysis of the expression of the truncated *O*-glycans Tn antigen and sialyl-Tn Antigen showed no differences, suggesting there is no general difference in the length of *O*-glycans between the *Fucosylated* and *Basal* subtypes (Fig. 5d). GALNT3 is a target of the family of ZEB transcription factors and its loss in poorly differentiated pancreatic cancer has been associated with a more aggressive phenotype[29]. We confirmed the expression of GALNT3 in *Fucosylated* cells by western blot (Fig. 5c) and confocal microscopy (Fig. 5e), the latter also showing a Golgi-like staining pattern. Moreover, we confirmed the expression of GALNT3 in tissue from patients with the *Fucosylated* subtype, while it was absent in the *Basal* subtype (Fig. 5f).

To investigate whether GALNT3 contributes to the *Fucosylated* phenotype, we generated knockout clones in the cell line PaTuS using the CRISPR-Cas9 technology and single-cell sorting and performed profiling of glycan structures using plant lectins and

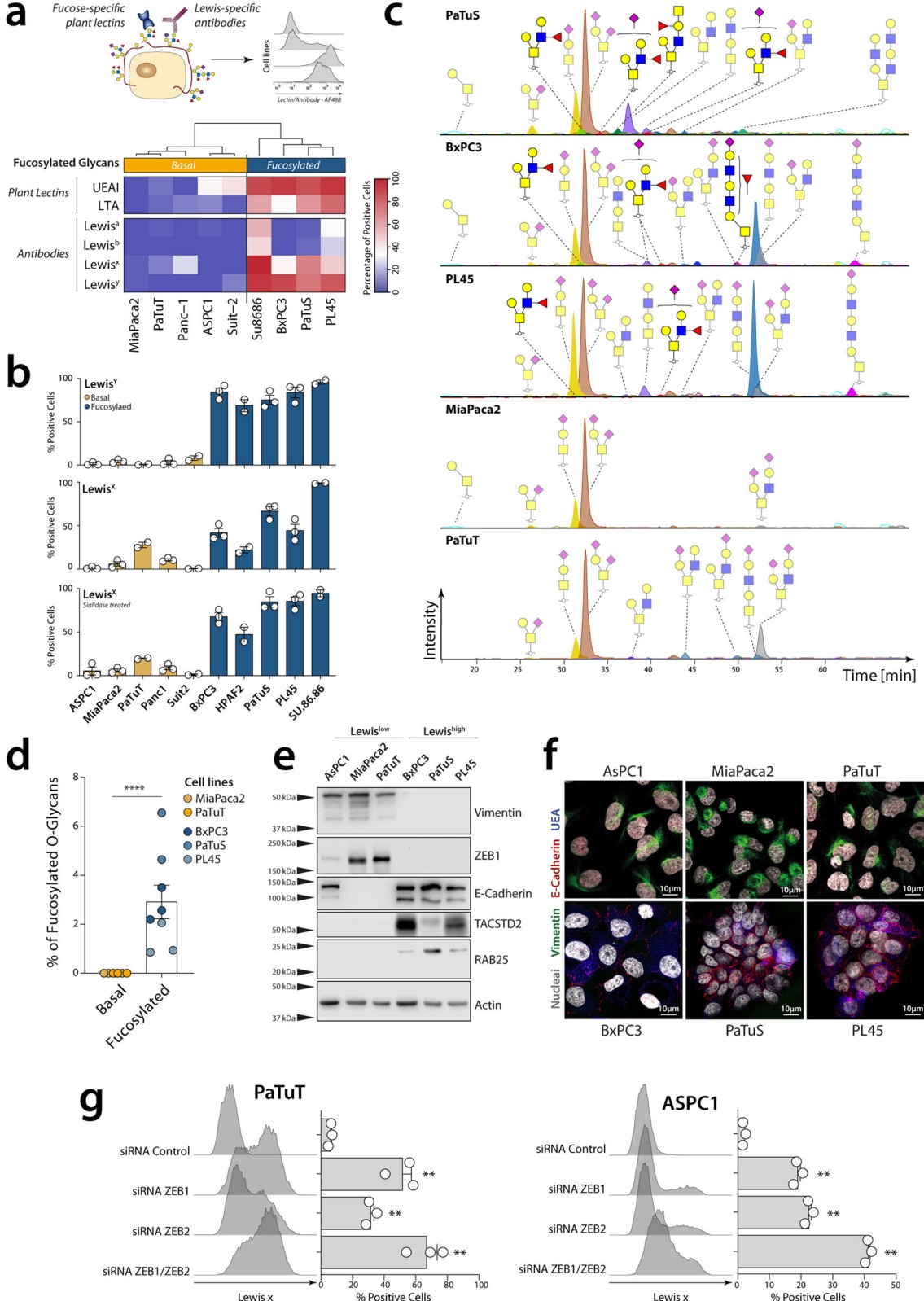

**Fig. 4 *Fucosylated* and *Basal* subtypes are also reflected in pancreatic cancer cell lines. a** Analysis of the profile of glycan structures expressed in cell lines using plant lectins or glycan-specific antibodies. **b** Quantification of the expression of Lewis[X] and Lewis[Y] in *Fucosylated* and *Basal* cell lines using anti-Lewis antigens antibodies, before and after sialidase treatment. Data presented as mean values ± SEM. **c** Chromatographic separation of *O*-glycans from PDAC cell lines. **d** Quantification of Fucosylated *O*-glyans present in cell lines. Statistics: Mann–Whitney test (*$p \leq 0.05$, **$p \leq 0.01$, ***$p \leq 0.001$, ****$p \leq 0.0001$). **e** Western blot and **f** confocal microscopy showing the differential expression of epithelial or mesenchymal markers in each subtype. **g** Knocking down of ZEB1 and ZEB2 in PaTuT and ASPC1 and evaluation of Lewis[X] expression by flow cytometry using specific antibodies. Data presented as mean values ± SEM. Statistics: using one-way ANOVA with Dunnett method for multiple test correction (**$p \leq 0.01$).

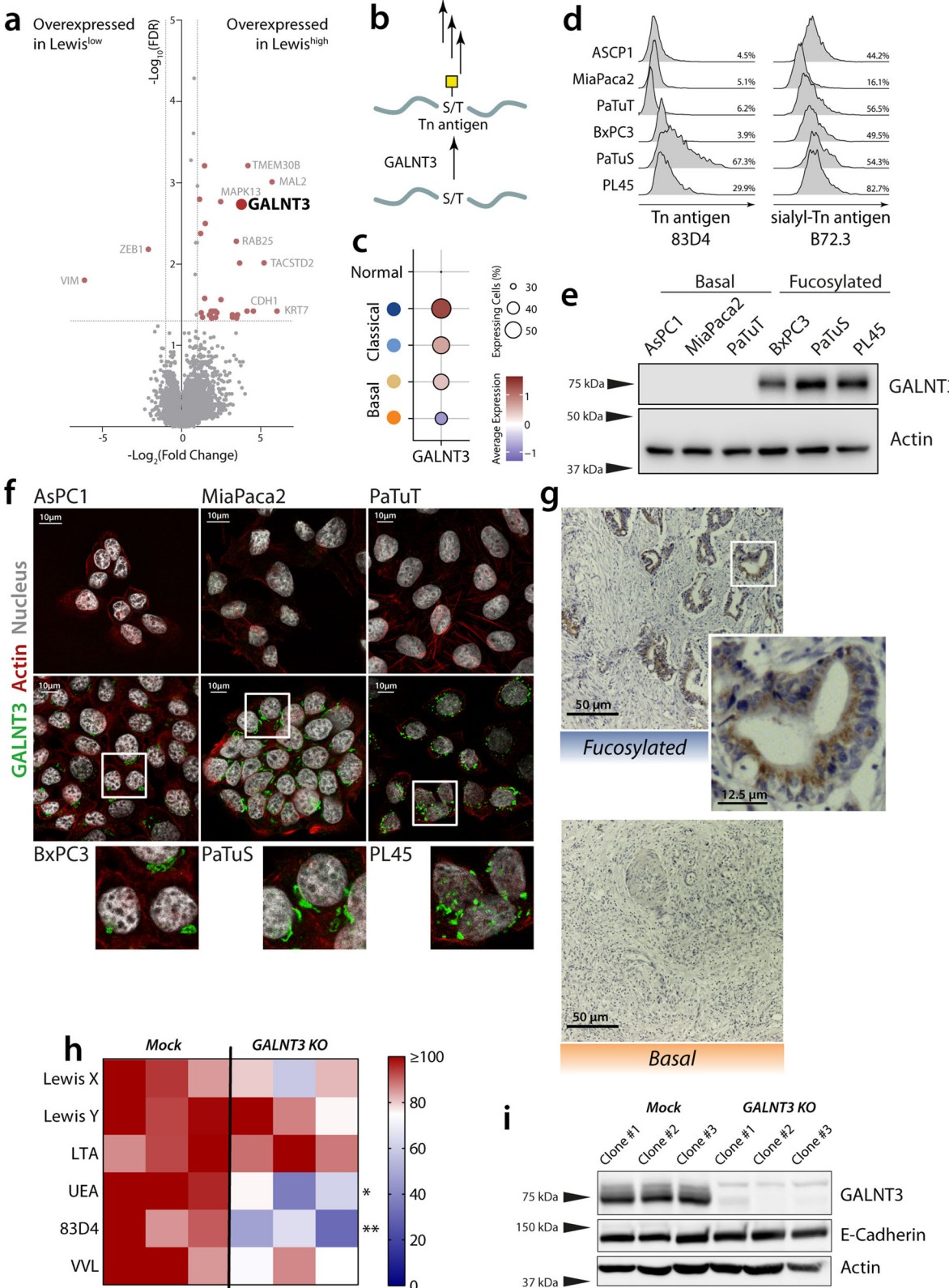

**Fig. 5 GALNT3 is overexpressed in *Fucosylated* cell lines and contributes to its glyco-phenotype. a** Volcano plot of a differential gene expressed genes between the Lewis[high] and Lewis[low] cell lines defined in. **b** Scheme of the GALNT3 enzymatic activity. **c** Expression of GALNT3 in the different tumour clusters in scRNA-seq. **d** Expression of Tn antigen (mAb: 83D4) and Sialyl-Tn antigen (mAb: B72.3) was analyzed by flow cytometry. **e** Western blot of GALNT3 in selected PDAC cell lines. **f** Confocal microscopy of GALNT3 (green) and actin (red). **g** Expression of GALNT3 is associated with the Fucosylated subtype in PDAC tissue. **h** Analysis of the expression of glycan structures using plant lectins and antibodies by flow cytometry in WT and GALNT3 KO cell lines. Statistical analysis: Two-way ANOVA with Sidak method for multiple test correction (*$p \leq 0.05$; **$p \leq 0.01$; ***$p \leq 0.001$). **i** Western Blot of GALNT3, E-Cadherin and Actin in WT and GALNT3 KO cell lines.

antibodies. GALNT3 KO clones had a lower expression of fucosylated glycans and Tn antigen (Fig. 5g). Interestingly, we did not see differences in the expression of the epithelial marker E-Cadherin (Fig. 5h) and or the mesenchymal marker Vimentin, as reported before[29].

**Glycan structures in PDAC cells mediate the interaction with TAMs.** Given that glycosylated structures can also serve as ligands for a vast array of immune lectin receptors, we investigated whether the specific glycan signatures of epithelial and mesenchymal cells also affect their recognition by the immune system. We focused on the myeloid cell-associated receptors DC-SIGN (which recognises fucosylated or poly-mannosylated glycans) and MGL (specific for structures with terminal GalNAc, as the Tn Antigen)[8]. Interestingly, ligands for DC-SIGN were present in epithelial-like cells, but absent in mesenchymal cells, while ligands of MGL could be found in both subtypes (Fig. 6a). CD163+ DC-SIGN+ cells could also be found in tumour tissue, indicating that tumour-associated macrophages (TAMs) from patients can express DC-SIGN and distinguish between Fucosylated and Basal tumour cells. (Fig. 6b, c). TAMs have been associated with the progression of different types of cancer and they correlate with poor survival in pancreatic cancer[13,16,30,31].

The triggering of DC-SIGN and MGL has been associated with the induction of tolerogenic circuits in dendritic cells, in particular by modulating the maturation by different TLR ligands[32,33]. Given that M2 macrophages can also express DC-SIGN and MGL, we continued to study whether glycan-mediated triggering of these receptors was able to modulate macrophage activation. For this aim, we used dendrimers carrying the Lewis$^x$ antigen or a terminal GalNAc to trigger the DC-SIGN and MGL receptors, respectively, in presence of or absence of the TLR ligands LPS (TLR4) and Pam3CSK4 (TLR2)[34]. Triggering of DC-SIGN increased the TLR-induced production of IL-10 and reduced IL-6, while glycan binding of MGL only induced a difference in IL-10 and TNFα (Fig. 6d). Interestingly, the combined triggering of both lectin receptors by both Lewis$^x$ and GalNAc synergised in their effect of reducing IL-6 production (Fig. 6c). To study if the triggering of DC-SIGN by tumour fucosylation can modulate macrophages in tumour tissue, we analysed scRNA-seq from PDAC tissue (Fig. 6e). For this, we evaluated the general expression of fucosylation-related genes detected in tumour cells using gene sets associated with the GDP-Fuc synthesis (*GMDS, TSTA3*), fucosyltransferases (*FUT2, FUT3, FUT4, FUT6*) or both (Figs. 3f, 6e). For each patient, the average expression of scores generated with these gene sets was correlated with the average expression of different cytokines by Macrophages, as described previously by our group[16]. Interestingly, we found that the tumour expression of fucosylation-related genes is negatively correlated with pro-inflammatory cytokines IL-8 and IL-6 in macrophages, while showing a positive correlation with TGF-β. Our data suggest that the Fucose-DC-SIGN axis contributes to the tolerogenic properties of macrophages in situ.

Given that GALNT3 KO cells present a decrease in the expression of some fucosylated antigens and that its expression was associated in epithelial cells, we analysed whether it could also contribute to the recognition of tumour cells by DC-SIGN. Indeed, knockout of GALNT3 in epithelial cells resulted in lower expression of DC-SIGN ligands (Fig. 6f). To study if the differential expression of DC-SIGN ligands in the cell line impacts macrophage activation, we co-cultured different cell lines with M2 macrophages in the presence of LPS. We observed an increased production of IL-10 when macrophages where co-cultured with the fucosylated cell lines PaTuS and PL45 (Fig. 6f).

Interestingly, the increase of IL-10 production was lost in the GALNT3 KO in PaTuS (Fig. 6g).

We, therefore, concluded that stimulation of TAMs by fucosylated antigens present in epithelial cells may contribute to the tolerogenic microenvironment during the early stages of PDAC.

## Discussion

Changes in glycosylated structures during malignant transformation have been widely reported in the literature over the last decades, highlighting the role of a diverse array of glycans in cancer progression depending on the tumour type, stage or grading[8]. This is also the case in PDAC, where different glycan structures or glycoproteins are used for diagnostics or have been proposed as targets in therapeutic approaches[10,35,36]. Here we describe, a deep characterisation of the tumour glyco-code in pancreatic cancer and its impact on immune-cancer interactions by using a multidisciplinary approach that involved transcriptomic analysis, glycoprofiling using mass spectrometry, lectin and antibody stanings, and functional assays.

The analysis of transcriptomic data of datasets of tissue samples from patients, and the subsequent validation in PDAC tissues, organoids and cell lines, allowed us to identify two glyco-code dependent subtypes of cancer cells (denominated as *Fucosylated* and *Basal*), that can be discriminated by the presence of fucosylated antigens. Each of these glycan signatures is associated with different extremes in the EMT continuum, with fucosylated structures being expressed in cells with an epithelial phenotype, while absent in mesenchymal-like cells. EMT has been considered a major driver of metastasis in PDAC, in a process dependent on the transcription factor ZEB1, but independent of SNAIL or TWIST[21,37]. The deletion of ZEB1 in animal models of pancreatic cancer led to the loss of cell plasticity and fixed tumour cells in an epithelial phenotype[21]. Interestingly, a recently reported glycomics analysis of colorectal cell lines showed that Lewis-type fucosylated *O*-glycans are restricted to differentiated tumour cells while are lost in undifferentiated tumour cells, in line with our results in PDAC[28].

We also found that ZEB1 was essential for the repression of fucosylated antigens in mesenchymal cell lines. Several genes can be repressed by ZEB1, including GALNT3, an enzyme overexpressed in epithelial cells and found to be associated with fucosylated antigens. GALNT3 has been shown to contribute to the phenotype of epithelial cells, with its downregulation being able to directly induce EMT[29,38]. Despite we couldn't see changes in the EMT status in PaTuS GALNT3 KO cell lines, the observed decreased expression of fucosylation antigens and DC-SIGN ligands (markers of epithelial PDAC cell lines) suggest that biological changes may be induced that are able to change the cell glycosylation but not changes in the expression of Vimentin or E-Cadherin in this particular cell line. However, the exact mechanisms behind the GALNT3-mediated control of fucosylation are still to be studied. Using an elegant *O*-glycoproteomic strategy, Narimatsu *et al* have identified a series of specific substrates for different GALNTs, including GLANT3[39]. Similar studies should be performed in PDAC epithelial cells to identify potential mechanisms by which GLANT3 regulates fucosylation. Moreover, our finding that the presence of GALNT3 can discriminate basal from epithelial (fucosylated) subtypes using immunohistochemistry, needs future investigations for its potential to be used as a diagnostic marker for subtype identification and patient stratification in PDAC.

Our results also show that the tumour glyco-code is heavily defined by cancer cells, with low purity samples clustering together in Cluster C. This cluster demonstrated no specific glycan

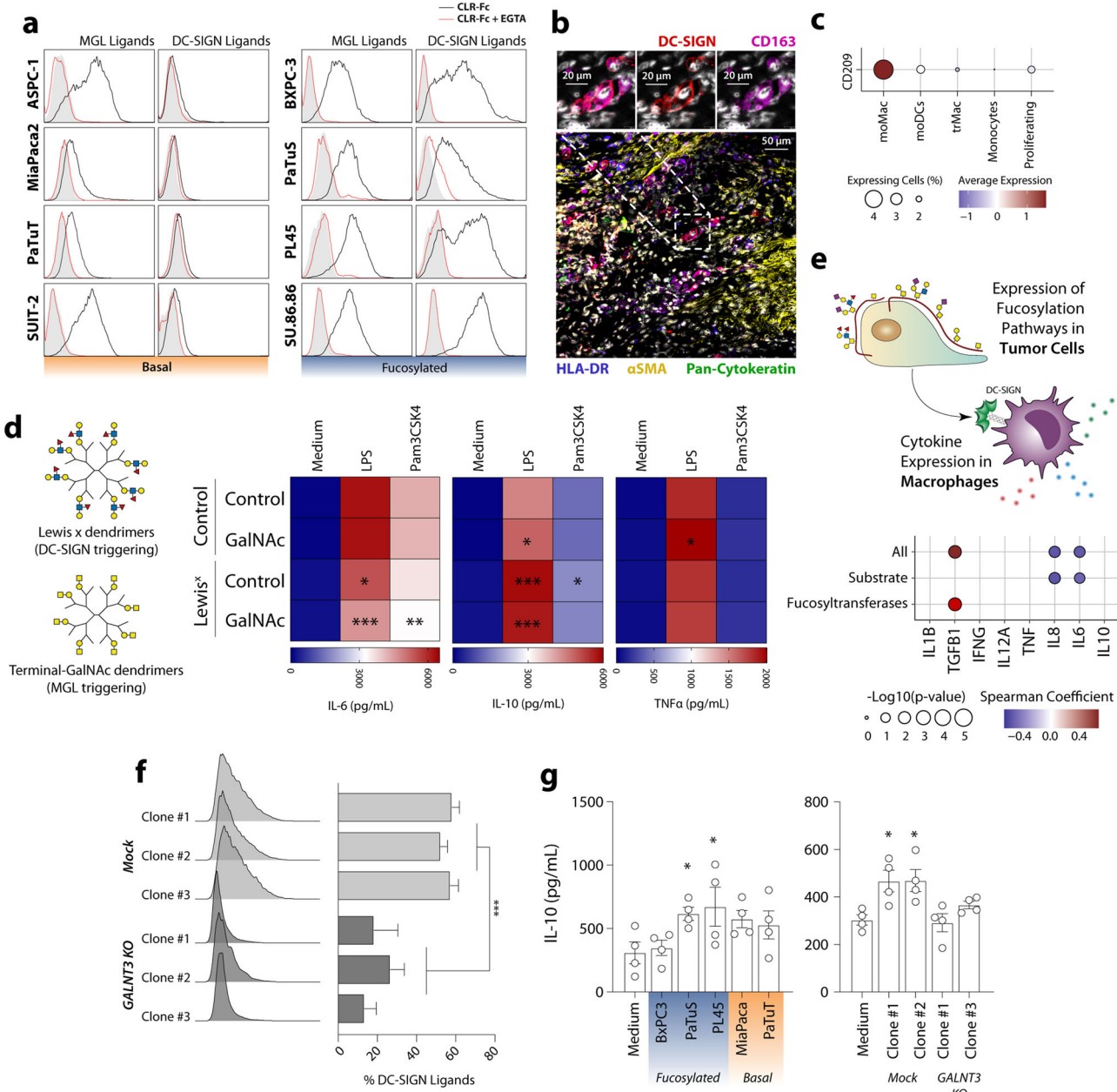

**Fig. 6 TAMs distinguish *Fucosylated* and *Basal* cells. a** Glycan ligands for the lectin receptors DC-SIGN and MGL were assessed by flow cytometry using DC-SIGNFc and MGL-Fc. **b** Confocal microscopy of tissue from PDAC patients identifying CD163[+] DC-SIGN[+] cells. **c** DC-SIGN (gene *CD209*) can be detected in moMac in scRNA-seq data of PDAC patients. **d** Stimulation of moMac macrophages ($n = 7$) with dendrimers containing Lewis[X] or terminal GalNAc in the presence or absence of TLR ligands, leads to altered IL-6 and IL-10 production, detected using ELISA. Statistics: comparisons of each condition against the stimulation with dendrimer control was performed by two-way ANOVA with Dunnett's multiple comparisons test (*$p \leq 0.05$; **$p \leq 0.01$; ***$p \leq 0.001$). **e** Correlation between the expression of Fucosylation pathway in tumour cells with the expression of cytokines in moMac using scRNA-seq. **f** Flowcytometric analysis of the expression of DC-SIGN ligands in WT and GALNT3 KO cell lines, using DC-SIGN-Fc. Statistical analysis: Two-way ANOVA (*$p \leq 0.05$, **$p \leq 0.01$ and ***$p \leq 0.001$). **g** IL-10 production of M2 Macrophages co-cultured with WT and GALNT3 KO cell lines in the presence of LPS, measured by ELISA. Statistics: Friedman test (*$p < 0.05$).

signature, which could be due to a high stromal and low tumour cell content. Future studies must be carried out to determine the role of the stromal glyco-code in cancer progression. This is particularly important in PDAC, which is characterised by a big desmoplastic reaction and the presence of different stromal cells[4]. Indeed, some studies have already described a differential expression of glyco-code related genes between tumour and stroma[40]. For example, while the stroma is enriched in the expression of Galectin-1, Galectin-4 is expressed mainly by tumour cells[40]. Stromal-derived galectin-1, can even contribute to

pancreatic cancer progression and represents an interesting target for anti-tumour therapies[41].

We here also describe how the different glyco-signatures of tumour cells during pancreatic cancer progression shape the immune cell–tumour interaction. The presence of fucosylated antigens in epithelial cells can serve as ligands for the tolerogenic receptor DC-SIGN, which is expressed in tumour-induced TAMs. Interestingly, the glycan-binding and triggering of DC-SIGN can clearly modulate the maturation of macrophages, by increasing the production of IL-10 and reducing the one of IL-6, which may

contribute to the tolerogenic microenvironment. Our data also show a synergistic effect between the simultaneous triggering of DC-SIGN and MGL, which deepens the increased production of IL-10 and the reduction in IL-6. Glycan ligands of MGL can be found on both ends of the EMT continuum, suggesting that this synergistic effect may take place in the early stages of PDAC development, thereby strongly contributing to the tolerogenic microenvironment, while the triggering of MGL can also help the immune scape of migrated mesenchymal cells[32]. The interaction of TAMs with tumour cells can also induce EMT in different types of cancer[42,43]. Given that DC-SIGN-expressing macrophages can differentially interact with the *Fucosylated* and *Basal* subtypes, the DC-SIGN-Fucose axis may mediate the first interactions with epithelial cells that later undergo EMT and lose DC-SIGN binding fucose.

In summary, we describe glycan-mediated tolerogenic circuits involved in PDAC and the induction of its tolerogenic microenvironment. Notably, our results should prompt new therapeutic opportunities to modulate the glyco-code, with emphasis on the inhibition of fucosylated structures for the future design of targeted strategies. The analysis of changes in the tumour glyco-code in other types of cancer will also reveal whether the mechanisms described here also apply to malignancies in other tissues or whether this is specific for pancreatic cancer.

## Methods

**PDAC patient tissue**. For the validation cohort (used in Figs. 2, 5), tissue of patients who underwent a pancreaticoduodenectomy (PD) for a PDAC at the Amsterdam UMC, location Academic Medical Centre Amsterdam (AMC) was used. The retrospective collection was conducted in accordance with ethical guidelines 'Code for Proper Secondary Use of Human Tissue in The Netherlands' (Dutch Federation of Medical Scientific Societies), approved by the Academic Medical Center's institutional review board (Medisch Ethische Toetsingscommissie AMC) under METC_A1 15.0122.

For the staining of DC-SIGN in tumour tissue (Fig. 6), we used specimens that were obtained from patients undergoing resection at the Amsterdam UMC, location VU University Medical Center, with the approval from the Local Medical Ethical committee at the VU and the Biobank (#14038). Written informed consent was obtained from each patient.

**Cell lines**. ASPC1, Mia PaCa-2 and PL45 were acquired from ATCC. BxPC3 are a kind gift from Dr. A. Frampton (Imperial College, London, UK). PaTuS and PaTuT were a kind gift from Dr. I. van Die (Amsterdam UMC, The Netherlands). Cell lines were tested for their authentication by STR-PCR, performed by BaseClear (Leiden, The Netherlands), previous to the start of the project. Additionally, all cell lines were routinely tested for Mycoplasma using PCR. All the cell lines were cultured in RPMI 1640 (Gibco) supplemented with 10% Foetal Calf Serum (Lonza), 2 mM L-glutamine (Gibco) and 1000 U/mL Penicillin-Streptomycin (Gibco).

**Transcriptomic analysis**. Data processing and analysis was performed in R language (v3.5.1) using RStudio. Characteristics of the datasets used in this paper (number of patients, platform, accession number and original publications) are detailed in Table S1.

Data deposited in the Gene Expression Omnibus (https://www.ncbi.nlm.nih.gov/geo/) was downloaded using the function *getGEO* of the R package *GEOquery*. Transcriptomic and clinical data (Release 27) from International Cancer Genome Consortium (ICGC) was downloaded from the ICGC data portal (https://dcc.icgc.org/). For the IGCG-PACA-AU project samples with the histological type described as 'Pancreatic Ductal Adenocarcinoma' were retained for further analysis. For ICGC-PACA-CA, samples from the primary tumour or metastasis and described as 'Enriched by Laser Capture Microdissection'. Data from the pancreatic cancer project of The Cancer Genome Atlas (TCGA) was through the Broad Institute TCGA firehose and the 150 samples described in Raphael et al.[14] were selected. RNA-seq data from cell lines was obtained from http://research-pub.gene.com/KlijnEtAl2014/.

The package *limma* was used for the analysis of the differential gene expression, with FDR correction for multiple comparisons. To identify clinically relevant glyco-code-related genes in pancreatic cancer, analysis between normal and tumour tissue was performed in paired (GSE15471 and GSE62452) or unpaired (GSE16515 and GSE71729) samples. Differential gene expression between clusters was performed using in pairwise comparisons of one cluster versus the others.

Genes differentially expressed between tumour and normal tissue in at least two datasets (cut off: *p* value ≤ 0.01) were selected for their use in consensus clustering using the package ConsensusClusterPlus[44]. Data were median centred and partitioning around medoids (PAM) was used as a clustering algorithm, using the

following conditions: 2000 iterations, 90% resampling, 1-Pearson's correlation as the distance metric and 'average' as linkage.

Kaplan–Meier curves for the analysis of the overall survival were generated using the function *ggsurvplot* (from the R package *survminer*) and the log-rank test was used to assess statistical significance.

To study the relation between the cluster defined in this paper with previously reported classification PDAC patients, we used a network-based analysis as described previously[45]. *ConsensusClusterPlus* was used for the classicisation of the discovery datasets in the different molecular subtypes using the gene classifier described by the authors[11–13]. In this network analysis, nodes represent the different molecular subtypes and edges represent their association as calculated by the Jaccard coefficient. Statistical significance was calculated by a hypergeometric test for overrepresentation and adjusted for multiple testing using the BH method. Significant edges ($p < 0.05$) were observed as network using *Cytoscape* and nodes were grouped using MCL clustering.

To study the association of each cluster with specific glycan signatures or previously reported subtypes, we performed gene set enrichment analysis on a single-sample basis using the *GSVA* package[46], using custom or published gene sets as follows:

- For the analysis of different glycosylation pathways, gene sets were used according to the division shown in Table S1.
- An 'O-Fucosylation' gene set was added, that include the genes: *LFNG, MFNG, RFNG, B3GLCT, B4GALT1, ST6GAL1, POFUT2* and *POFUT1*.
- Gene sets that represent each or the previously reported PDAC subtypes was obtained from the respective publications[11–13].
- An 'EMT Score' was obtained as the difference of an epithelial and a mesenchymal score, using gene sets described previously[47].

Statistical significance between clusters for a given gene set was determined by the Kruskal–Wallis test, analyzing pairwise comparisons of one cluster versus the others using the function *kruskal.test*. Bubble plots were generated using the R package *ggplot2*, using the GSVA score to determine the colour and the $-\log_{10}$ of the *p* value for the size.

For the characterisation of the *Fucosylated* and *Basal* subtypes found in laser microdissected data and cell lines we performed Gene Set Enrichment Analysis (GSEA) as implemented in the *javaGSEA Desktop Application* and visualised in *Cytoscape* using the *EnrichmentMap* app. Kyoto Encyclopaedia of Genes and Genomes (KEGG), REACTOME and HALLMARK gene sets were obtained from the Molecular Signatures Database (MSigDB) from the Broad Institute web page (http://software.broadinstitute.org/gsea/msigdb).

The scRNA-seq data previously published by Peng et al. was downloaded from the Genome Sequence Archive project PRJCA001063 as preprocessed row data and imported into the package *Seurat* (v3) for downstream analysis[25,48]. The *SCTransform* function was used to normalise and scale the data, regressing out the mitochondrial percentage and Principal Component Analysis (PCA) was performed with the 3000 most variable genes. PCA components 1 to 10 were used for graphical-based clustering at a resolution of 1. Four clusters that identified by Peng et al.[25]. These clusters were projected onto Uniform Manifold Approximation and Projection (UMAP) dimensional reduction. Scores for EMT and the Classical and Basal subtypes defined by Moffit et al. were performed using the function *AddModuleScore*[12].

**Glycan profiling by flow cytometry**. All the staining with Plant lectins (Vector Laboratories) or chimeric receptors (in house) were performed in 10% BSA in HBSS supplemented with calcium and magnesium (Gibco). A total of 100,000 cells were incubated with 2 μg/mL of biotinylated lectin or 10 μg/mL Fc-chimeras for 45 min at 4 °C. The detection was performed by incubating the cells with Alexa-Fluor 488-conjugated Streptavidin (Invitrogen) or FITC-conjugated anti-human IgG Fc (Jackson ImmunoResearch). To evaluate whether fucosylated structures are present in N-glycans, O-glycans or Glycolipids, cells were treated for 3 days with 2 mM Benzyl-GalNAc, 10 μg/mL Kifunensine or 1 mM PPMP, respectively.

**Analysis and quantification of O-glycans**. O-Glycans present in PDAC cell lines were analysed as described before[28]. Briefly, lysed cell pellets containing 500,000 cells were loaded to preconditioned PVDF membrane plate (Millipore) wells and denatured with guanidine hydrochloride and dithiothreitol (DTT). After removing the denaturation agent, N-glycans were released by PNGase F (Roche) digestion overnight at 37 °C. Upon removal of N-glycans, 50 μL of 0.5 M sodium borohydride in 50 mM potassium hydroxide was added to each well and incubated for 16 h at 50 °C for the release of O-glycans via reductive beta-elimination. Upon recovery of the released O-glycans, samples were desalted by cation exchange solid-phase extraction (Dowex (50W-X8), Sigma- Aldrich). Desalted O-glycans were further purified via solid-phase extraction by packing bulk sorbent PGC slurry (Grace Discovery sciences) into 96-well filter plates. Analysis was performed using a PGC nano-LC-ESI-MS/MS platform. Identification of glycans was performed based on PGC retention time, known biosynthetic pathways, and manual inspection of MS/MS spectra following known fragmentation pathways of O-glycan alditols in negative-ion mode. Relative quantitation was performed on the total area of all O-glycans within one sample normalising it to 100%.

**Table 1 List of Antibodies used in this paper.**

| Antibody | Source | Catalogue No. | Dilution |
|---|---|---|---|
| Flow Cytometry and Microscopy | | | |
| JoJo1 | Invitrogen | J11372 | 1:15000 |
| Anti-PanCytokeratine - AlexaFluor 488 | eBioscience | 53-9003-82 | 1:100 |
| Anti-CD163 - Brilliant Violet 421™ | Biolegend | 333612 | 1:50 |
| Anti-Alpha-Smooth Muscle Actin - eFluor® 570 | eBioscience | 41-9760-82 | 1:200 |
| Anti-DC-SIGN (AZN-D1) - AlexaFluor 488 | In house. | | 1:50 |
| Anti-HLA-DR - BV786 | BD Biosciences | 564041 | 1:100 |
| Anti-E-Cadherin - AlexaFluor 647 | Biolegend | 147307 | 1:250 |
| Anti-Vimentin - AlexaFluor 488 | Biolegend | 677809 | 1:500 |
| Goat anti-rabbit IgG - AlexaFluor 488 | Invitrogen | 11008 | 1:500 |
| Anti-Lewis$^Y$ antibody | GeneTex | GTX23359 | 1:100 |
| Anti-Lewis$^X$ antibody | CalBiochem | 434631 | 1:25 |
| Anti-mouse IgM - FITC | Jackson ImmunoResearch | 115-096-075 | 1:100 |
| Anti-mouse IgG - FITC | Jackson ImmunoResearch | 115-096-072 | 1:50 |
| Anti-human IgG - FITC | Jackson ImmunoResearch | 109-096-098 | 1:50 |
| Western Blot | | | |
| CA19-9 Antibody (SPM110) | Thermo Fisher | MA5-14383 | 1:500 |
| Purified anti-human E-Cadherin (24E10) | Cell Signaling | 3195 S | 1:1000 |
| Purified anti-human Vimentin | Biolegend | 677802 | 1:500 |
| Purified anti-human TMEM30B | Novus Biologicals | NBP1-59534 | 1:500 |
| Purified anti-human MAL2 | Abcam | ab75347 | 1:1000 |
| Purified anti-human RAB25 | Cell Signaling | 13048 T | 1:500 |
| Purified anti-human GALNT3 Antibody | R&D Systems | AF7174-SP | 1:200 |
| Polyclonal anti-human ZEB1 | Atlas Antibodies | HPA027524 | 1:100 |
| Purified anti-β-actin Antibody | Biolegend | 643802 | 1:1000 |
| Goat anti-rabbit - HRP | Dako | P0448 | 1:2000 |
| Rabbit anti-mouse - HRP | Dako | P0161 | 1:1500 |
| Goat anti-sheep - HRP | R&D Systems | HAF016 | 1:1000 |
| ELISA | | | |
| Capture Antibody - IL-10 | eBioscience | 14-7108-85 | 1:2000 |
| Detection Antibody - IL-10 | eBioscience | 13-7109-85 | 1:2000 |
| Capture Antibody - IL-6 | Biosource | AHC0562 | 1:2000 |
| Detection Antibody - IL-6 | Biosource | AHC0469 | 1:2500 |
| Capture Antibody - TNFα | Biosource | AHC3712 | 1:1000 |
| Detection Antibody - TNFα | Biosource | AHC3419 | 1:1000 |

**Western blot**. Lysates were obtained by extensive washing with ice-cold PBS and incubation in the culture flask with RIPA buffer (50 mM Tris-HCl pH = 8, 150 mM NaCl, 2 mM EDTA, 1% NP-40, 0.5% sodium deoxycholate and 0.1% SDS) supplemented with protease inhibitors (Roche). Trypsinization of the cells was avoided to preserve epitopes. Protein concentration was performed by the bicinchoninic acid (BCA) assay (Pierce). About 10 μg of protein were separated by SDS-PAGE and transferred onto nitrocellulose membranes. The membranes were with 1% BSA in PBS and then incubated with anti-human E-Cadherin (24E10), anti-human Vimentin, anti-human ZEB1, anti-human TMEM30B, anti-human Rab25 or anti-human GALNT3 antibodies overnight at 4 °C. HRP-conjugated anti-mouse (Vimentin), anti-rabbit (E-Cadherin, RAB25, TMEM30B) or anti-sheep (GALNT3) antibodies were used as secondaries. The details of the antibodies used in this paper are shown in Table 1. Complete western blot are presented in Fig. S7.

**Microscopy**. From formalin-fixed paraffin-embedded (FFPE) PDAC tissue blocks acquired during routine patient care in the Amsterdam University Medical Center, an haematoxylin-eosin (HE)-stained section was assessed by a pathologist, specialised in pancreas pathology, to identify an area with invasive PDAC. From each tumour two 2-mm-cores were collected using the Beecher TMA instrument and inserted into a recipient block using the Manual Tissue Arrayer MTC-1 (Beecher Instruments) to construct the TMAs. Immunohistochemical staining for Mucin 16 (MUC16 or CA125; monoclonal antibody Thermo Scientific MS-1812-S Clone MII; diluted 1:250) was performed on 4 μm sections of the TMA blocks and carried out using the Ventana Benchmark XT automatic staining system (Ventana Medical Systems).

For the phenotype characterisation of cell lines, 25,000 cells were seeded in μ-Slide 8 Well (IBIDI) and cultured overnight. For the analysis of EMT marker expression, cells were washed twice with 10% BSA in HBSS (Gibco), incubated with UEA I lectin (2 μg/mL) at room temperature for 2 h and fixated with 2% paraformaldehyde in HBSS. Permeabilization of the cells was performed with 0.1% Triton® X-100 for 15 min, washed and blocked with 4% FCS (Lonza). Cells were then stained with AlexaFluor 488-conjugated anti-Vimentin (Biolegend), AlexaFluor 647-conjugated anti-E-Cadherin (Biolegend), AlexaFluor 555-conjugated Streptavidin (Thermo Fisher Scientific) and DAPI (Invitrogen). For the

detection of GALNT3, cells were fixed with 2% paraformaldehyde in PBS, permeabilized with 0.1% Triton® X-100, blocked with 4% FCS (Lonza) and incubated with Rabbit polyclonal antibody anti-human GALNT3 (Atlas Antibodies). Detection was performed using an AlexaFluor 488-conjugated anti-rabbit antibody. Cells were also counterstained with DAPI (Invitrogen) and AlexaFluor 598-conjugated Phalloidin. The details of the antibodies used in this paper are shown in Table 1.

**Glycodendrimer synthesis**. The generation 2.0 PAMAM dendrimer with a cystamine core (647829, Sigma Aldrich) was conjugated to three different glycans via reductive amination as shown in Fig. S8. 20 equivalents of α-D-N-acetylgalactosaminyl 1-3 galactose (G283, Dextra Laboratories UK), Lewis$^X$ tetraose (GLY050, Elicityl) and D-(+)-Galactose (G0750 Sigma Aldrich) per dendrimer were dissolved in Dimethylsulphoxide (DMSO) and acetic acid (8:2). Per dendrimer, 160 equivalents 2-Methylpyridine borane complex (65421 Sigma Aldrich) was added to a total volume of 200 μL. The reaction was incubated at 65 °C for 2 h with frequent vortexing. The reaction products were purified over disposable PD10 desalting columns (GE17-0851-01 GE Healthcare) in 50 mM Ammonium Formate pH 4.4 (NH$_4$HCO$_3$). Multiple lyophilization cycles retrieved the glyco-dendrimers, which were later validated by LC-MS. Dendrimers containing terminal GalNAc were designed for the triggering of MGL (generated with α-D-N-acetylgalactosaminyl 1-3 galactose) and Lewis$^X$ dendrimers for the stimulation of DC-SIGN, while Galactose dendrimers were used as controls[34].

**Macrophage differentiation and stimulation**. Buffy coats were obtained from healthy donors (Sanquin, The Netherlands). Peripheral blood mononuclear cells (PBMCs) were isolated by density gradient centrifugation with Ficoll-Paque PLUS (GE Healthcare), to purify CD14$^+$ monocytes using MACS CD14 MicroBeads (Miltenyi).

For the stimulation with glyco-dendrimers, moMacs were generated by incubating monocytes with 50 ng/mL M-CSF for 5 days, with a change at day 3. Polarisation towards M2 macrophages was performed by stimulating with 20 ng/mL IL-4 and 20 ng/mL IL-6 for 2 days. Given that MGL triggering

required immobilised (not soluble) ligands[32], 1 μM of the control and terminal GalNAc-dendrimers were coated in sterile 96-wells NUNC plates overnight at room temperature. About 50,000 moMacs were added to the plate in the presence of 1 μM of Lewis[X] or control dendrimers in the presence or absence of 10 ng/mL LPS (Sigma Aldrich) or 10 μg/mL Pam3CSK4. Supernatants were harvested 18 h later for the determination of cytokine secretion by ELISA. The antibodies used are shown in Table 1.

**ZEB1 and ZEB2 knockdown.** To study the role of the transcription factor ZEB1 and ZEB2, we knocked down their expression by using SMARTpools (Dharmacon), consisting of four different small interfering RNA (siRNA) that target either ZEB1 or ZEB2. ASPC1 and PaTuT were transfected with SMARTpools targeting ZEB1 and ZEB2 using DharmaFECT 2 Transfection Reagent (Dharmacon). After 3 days, cells were analyzed for Lewis[X] expression by flow cytometry. As a control, a nontargeting siRNA pool was used (Dharmacon).

**CRISPR-Cas9 gene knockout.** The generation of GALNT3 knockout clones of the cell line PaTuS was performed as previously described[49]. Briefly, cells were transfected with pSpCas9n(BB)-2A-GFP (Addgene plasmid ID: 48140) plasmids having guide RNAs for GALNT3, using Lipofectamine™ LTX with PLUS™ Reagent (Invitrogen). Single-cell sorting of the transfected cells based on GFP expression was performed using BD FACSAria™ Fusion FACS sorter. Individual clones were selected based on their expression of GALNT3 as analyzed by western blot. Mock cell lines were transfected with pSpCas9n(BB)-2A-GFP without guide RNA.

**Statistics and reproducibility.** For data derived from the transcriptomic analysis (Figs. 1–3 and Figs. S1–S5), the statistical analysis was performed in R (v3.5.1). GraphPad was used for the rest of the analysis. Specific tests performed are indicated in each figure. Log-rank test was performed to determine statistical significance in survival analysis, as implemented in the function *ggsurvplot* of the package *survminer*. All the graphs are represented as the mean ± standard error of the mean (SEM).

The data presented in this paper was generated in two or more independent experiments. Assays involving the stimulation of monocyte-derived macrophages were performed in at least four different donors.

**Reporting summary.** Further information on research design is available in the Nature Research Reporting Summary linked to this article.

## Data availability

The transcriptomic data that support the findings of this study are publicly available, as detailed in Table S1. The analysis of differential gene expression between normal and tumour tissue was obtained from the NCBI GEO database (https://www.ncbi.nlm.nih.gov/geo/) with the following accession numbers: GSE15471[50], GSE16515[51], GSE62452[52] and GSE71729[12]. The single-cell RNA-seq data from Peng et al. was downloaded from the Genome Sequence Archive (https://bigd.big.ac.cn/gsa/) under the project PRJCA001063[25]. The data from the PAAD project of the TCGA is available for download from the Broad Institute GDAC Firehose (https://gdac.broadinstitut.org)[14]. The transcriptomic data published by Puleo et al. was downloaded from the ArrayExpress database with the accession number E-MTAB-6134[40]. Data from the PACA-AU project of the ICGC (release 25) was obtained from https://dcc.icgc.org/[13]. The remaining data are available within the Article, Supplementary Data file or available from the authors upon request.

The plasmid pSpCas9n(BB)-2A-GFP, used in the generation of GALNT3 KO cells by CRISPR-Cas9, was obtained from Addgene (Plasmid ID: 48140).

## Code availability

The code for the transcriptomic analysis in this manuscript is available from the authors upon request.

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

## Acknowledgements

A significant amount of the results in this paper is based on the transcriptomic analysis of publicly available datasets. We would like to acknowledge the work of the research groups responsible for the generation and processing of the datasets used in this paper. We would like to thank Dr. Eduardo Osinaga and Dr. Teresa Freire (Facultad de Medicina, Universidad de la Republica, Uruguay) for their generosity in providing the antibodies 83D4 and B72.3. We also acknowledge the Microscopy and Cytometry Core Facility at the Amsterdam UMC (Location VUmc) for providing assistance in cytometry and microscopy experiments. E.R. is supported by Immunoshape (MSCA-ITN-2014-ETN No 642870) and Spinoza grant. S.T.T.S. and Y.v.K. are supported by the European Research Council (ERC-339977-Glycotreat). K.Bo. is supported by KWF VU2014-7200.

## Author contributions

E.R. and Y.v.K. conceived the study. E.R., K.Bo., K.M., F.D., J.V., R.J.E.L., S.T.T.S., L.L.M., T.Y.S.L.L., E.D., H.C., S.J.v.V., M.F.B., M.W., G.K., E.G., J.J.G.-V. and Y.v.K. designed experiments or provided essential technical support. E.R., K.Bo., K.Br., K.M., T.E., F.D., J.V., T.T. and S.B.acquired experimental data. E.R., K.Bo., K.Br., K.M. analyzed data. E.R. performed transcriptomic analyses. E.R and Y.K. drafted the manuscript. E.R., K.Bo., K.Br., K.M., L.L.M., T.Y.S.L.L., M.F.B., M.W., G.K., S.J.v.V., E.G., J.J.G.-V. and Y.v.K. provided critical intellectual content. Y.v.K. supervised the study.

## Competing interests

The authors declare no competing interests.
