## [Peer Review File · Communications Biology]

Reviewers' comments:

Reviewer #1 (Remarks to the Author):

The manuscript submitted by Rodriguez et al. is a revised study which has previously been separated into two distinct manuscripts. The authors have produced an impressive and comprehensive transcriptomic analysis of glycosylation associated gene profiles in multiple data sets, including bulk patient sequencing, sc-RNA sequencing, cell lines and organoids. Whilst this type of analysis relies heavily upon transcriptomics and is largely descriptive, the data sets generated are novel, and in my opinion are of great interest and importance to the wider research field.

In my opinion, the authors have done good job of starting to understand the complexities of glycosylation in the context of PDAC, and have demonstrated the utility of large transcriptomic datasets as a starting point for predicting changes at the glycan level. They have successfully validated some of their predicted glycosylation changes (ie, MUC16) successfully.

I agree that the wider field does not fully understand exactly how glycosylation enzymes work, however in-depth profiling studies such as this one make for a good starting point to better understand these complex processes. Regarding the observed changes in fucosylation in response to changes in GALNT3 expression, we have seen similar changes in our work on GALNT expression (unpublished data). In their rebuttal letter, the authors suggest that this could be through an indirect mechanism. I would suggest a potential explanation being that increased GALNT expression may result in increased GalNAc containing glycans which can then be extended by fucosylation. This result, to my mind, makes sense, and has been successfully demonstrated by the authors.

Changes

I think that the weakness of this manuscript is in its interrogation of DC-SIGN/MGL as functional effectors of the tolerogenic microenvironment. The authors successfully identify expression of DC-SIGN/MGL ligands in epithelial (DC-SIGN/MGL) and mesenchymal (MGL) cells in PDAC, and expression of DC-SIGN on TAM's, and show that loss of GALNT3, decreases DC-SIGN ligands, ameliorating IL-10 signalling. The authors state that they are studying 'glycan mediated triggering' of DC-SIGN. I don't think that there is strong experimentally proven evidence that DC-SIGN is being triggered in this instance. Could the authors demonstrate DC-SIGN 'triggering' by looking at its expression (either by western or IF) in response to Lewisx or GalNAc dendrimers. To further prove that this is a DC-SIGN dependant mechanisms could the authors look at targeting DC-SIGN (ie by siRNA knockdown) and measuring IL-10 and IL-6 outputs? If not, I would suggest that the authors alter the text so that not to over-state their findings (ie, line 362: We therefore concluded that triggering of DC-SIGN on TAMs by fucosylated antigens present in epithelial cells may contribute to the tolerogenic microenvironment during the early stages of PDAC).

Reviewer #2 (Remarks to the Author):

The authors have done a reasonable amount of changes for publication.

Reviewer #3 (Remarks to the Author):

The authors perform a thorough analysis of the glyco-code in pancreatic ductal adenocarcinoma (PDAC) with the goal to get a better insight into the glycan patterns governing the immune cells to produce a tolerogenic tumor microenvironment (TME) that contributes to tumor progression. The approach involves the use of publicly available transcriptomic data of patient samples and cell lines to identify glycan profiles using mass spectrometry and lectin staining. In addition, the authors display clinical relevance for their findings by studying the interaction between specific glycan signatures and tumor-associated macrophages (TAMs). Transcriptomic analysis revealed that

sialylation, O-glycosylation, and fucosylation are generally increased in PDAC. Furthermore, the authors carefully carried out a network and gene set enrichment analysis to group the specific glycan signatures to previously defined molecular subtypes identified in pancreatic cancer. Two main subtypes (clusters) were identified, the Fucosylated and the Basal, where the former comprises of genes involved in fucosylation and O-glycan extension and the latter for galectin-1 and mucins, MUC4 and MUC16. The two subtypes were further characterized based on their epithelial to mesenchymal transition (EMT), where higher EMT phenotype correlated to decrease in expression of O-glycosylation and fucosylation, and poor survival. Next, the authors clearly showed that GALNT3 was the main enzyme expressed in the Fucosylated subtype. Lastly, the authors used myeloid cell-associated receptor DC-SIGN and MGL on TAMs to demonstrate how they can differentially interact with the Fucosylated and Basal subtypes and contribute to the tolerogenic TME by regulating IL-10 and one of the IL-6.

Overall, this is a very comprehensive study to provide an account of the glycan signatures present in pancreatic cancer. The authors did a good job in verifying the feasibility of their defined molecular subtypes in cells and understanding their mechanism of interaction with immune receptors and ligands that contribute to PDAC.

The authors may want to consider the following comments as they continue with the publication process.

1. Transcriptomic analysis revealed that sialylation, O-glycosylation, and fucosylation are generally increased in PDAC.

Was there any differences among the different glycosylation genes? For example, was the expression of genes encoding for GDP-Fuc higher than genes for sialyl-Tn in the datasets?

2. The two subtypes were further characterized based on their epithelial to mesenchymal transition (EMT), where higher EMT phenotype correlated to decrease in expression of O-glycosylation and fucosylation, and poor survival.

Did the authors correlate genes of Cluster C subtype with Classical Score as a negative control for analysis of EMT in pancreatic cancer?

3. Regarding the identification of Fucosylated subtype by plant lectins, both lectins recognize alpha-L-fucose, however, there are some differences seen between the % of positive cells between UEAI and LTA, what can the authors attribute this difference to? Or is it insignificant? Is there a standard deviation (SD) reported?

4. The glycodendrimer synthesis can benefit from the inclusion of a scheme (in the Supplemental Materials section) and more info about the characterization of these dendrimers. Ref. 33 does not provide enough info either.

Reviewers' comments:

Reviewer #1 (Remarks to the Author):

The manuscript submitted by Rodriguez et al. is a revised study which has previously been separated into two distinct manuscripts. The authors have produced an impressive and comprehensive transcriptomic analysis of glycosylation associated gene profiles in multiple data sets, including bulk patient sequencing, sc-RNA sequencing, cell lines and organoids. Whilst this type of analysis relies heavily upon transcriptomics and is largely descriptive, the data sets generated are novel, and in my opinion are of great interest and importance to the wider research field.

In my opinion, the authors have done good job of starting to understand the complexities of glycosylation in the context of PDAC, and have demonstrated the utility of large transcriptomic datasets as a starting point for predicting changes at the glycan level. They have successfully validated some of their predicted glycosylation changes (ie, MUC16) successfully.

I agree that the wider field does not fully understand exactly how glycosylation enzymes work, however in-depth profiling studies such as this one make for a good starting point to better understand these complex processes. Regarding the observed changes in fucosylation in response to changes in GALNT3 expression, we have seen similar changes in our work on GALNT expression (unpublished data). In their rebuttal letter, the authors suggest that this could be through an indirect mechanism. I would suggest a potential explanation being that increased GALNT expression may result in increased GalNAc containing glycans which can then be extended by fucosylation. This result, to my mind, makes sense, and has been successfully demonstrated by the authors.

We would like to thank the reviewer for their positive comments. It is encouraging to us that the reviewer observes similar results that we do, about the control of fucosylation by GALNT enzymes. We also agree that our results could be explained by the fact that the expression of GALNT3 could increase the expression of O-Glycans that can be fucosylated.

Changes

I think that the weakness of this manuscript is in its interrogation of DC-SIGN/MGL as functional effectors of the tolerogenic microenvironment. The authors successfully identify expression of DC-SIGN/MGL ligands in epithelial (DC-SIGN/MGL) and mesenchymal (MGL) cells in PDAC, and expression of DC-SIGN on TAM's, and show that loss of GALNT3, decreases DC-SIGN ligands, ameliorating IL-10 signalling. The authors state that they are studying 'glycan mediated triggering' of DC-SIGN. I don't think that there is strong experimentally proven evidence that DC-SIGN is being triggered in this instance. Could the authors demonstrate DC-SIGN 'triggering' by looking at its expression (either by western or IF) in response to Lewisx or GalNAc dendrimers. To further prove that this is a DC-SIGN dependant mechanisms could the authors look at targeting DC-SIGN (ie by siRNA knockdown) and measuring IL-10 and IL-6 outputs? If not, I would suggest that the authors alter the text so that not to over-state their findings (ie, line 362: We therefore concluded that triggering of DC-SIGN on TAMs by fucosylated antigens present in epithelial cells may contribute to the tolerogenic microenvironment during the early stages of PDAC).

We agree with the reviewer we did not fully proof that the effect of fucosylated glycans in the modulation of macrophages depends on DC-SIGN signaling.

Using DC-SIGN-hFc chimeric receptors we showed that Lewis x dendrimers interact with DC-SIGN. Moreover, if we assess the expression of membrane DC-SIGN by FACS in Macrophages stimulated with dendrimers, we can see a strong reduction in surface DC-SIGN in macrophages stimulated with dendrimers containing Lewis x antigen but not in dendrimers Control (Figure). Given that DC-SIGN is an endocytic receptor, this may reflect its specific interaction of Lewis x dendrimers.

However, given that neutralizing experiments, in which we used blocking anti-DC-SIGN antibodies, were not successful in our hands, we have changed the line 362 for the following:

“We therefore concluded that stimulation of TAMs by fucosylated antigens present in epithelial cells may contribute to the tolerogenic microenvironment during the early stages of PDAC.”

Reviewer #2 (Remarks to the Author):

The authors have done a reasonable amount of changes for publication.

We would like to thank the reviewers for their kind comments.

Reviewer #3 (Remarks to the Author):

The authors perform a thorough analysis of the glyco-code in pancreatic ductal adenocarcinoma (PDAC) with the goal to get a better insight into the glycan patterns governing the immune cells to produce a tolerogenic tumor microenvironment (TME) that contributes to tumor progression. The approach involves the use of publicly available transcriptomic data of patient samples and cell lines to identify glycan profiles using mass spectrometry and lectin staining. In addition, the authors display clinical relevance for their findings by studying the interaction between specific glycan signatures and tumor-associated macrophages (TAMs). Transcriptomic analysis revealed that sialylation, O-glycosylation, and fucosylation are generally increased in PDAC. Furthermore, the authors carefully carried out a network and gene set enrichment analysis to group the specific glycan signatures to previously defined molecular subtypes identified in pancreatic cancer. Two main subtypes (clusters) were identified, the Fucosylated and the Basal, where the former comprises of genes involved in fucosylation and O-glycan extension and the latter for galectin-1 and mucins, MUC4 and MUC16. The two subtypes were further characterized based on their epithelial to mesenchymal transition (EMT), where higher EMT phenotype correlated to decrease in expression of O-glycosylation and fucosylation, and poor survival. Next, the authors clearly showed that GALNT3 was the main enzyme expressed in the Fucosylated subtype. Lastly, the authors used myeloid cell-associated receptor DC-SIGN and MGL on TAMs to demonstrate how they can differentially interact with the Fucosylated and Basal subtypes and contribute to the tolerogenic TME by regulating IL-10 and one of the IL-6.

Overall, this is a very comprehensive study to provide an account of the glycan signatures present in pancreatic cancer. The authors did a good job in verifying the feasibility of their defined molecular subtypes in cells and understanding their mechanism of interaction with immune receptors and ligands that contribute to PDAC.

The authors may want to consider the following comments as they continue with the publication process.

1. Transcriptomic analysis revealed that sialylation, O-glycosylation, and fucosylation are generally increased in PDAC. Was there any differences among the different glycosylation genes? For example, was the expression of genes encoding for GDP-Fuc higher than genes for sialyl-Tn in the datasets?

First, we would like to thank the reviewer for their comments.

In our analysis we did not observe clear differences among glycosylation genes.

We did observe that genes associated with different glycosylation pathways (as synthesis of both GDP-Fucose and Sialyl-Tn) are differentially expressed in PDAC when compared with normal pancreas. However, we did not observe clear patterns of expression depending on the glycosylation pathway, except for a general upregulation of genes involved in O-glycosylation, as stated in the paper.

Moreover, the diverse complexity of the different glycosylation pathways makes more difficult this analysis. For example, considering the pathways suggested by the reviewer: while the expression of *ST6GALNAC1* can be used to evaluate Sialyl-Tn antigen synthesis pathway, the analysis of expression of several genes are needed to evaluate the GDP-Fucose (Supplementary Figure 1).

2. The two subtypes were further characterized based on their epithelial to mesenchymal transition (EMT), where higher EMT phenotype correlated to decrease in expression of O-glycosylation and fucosylation, and poor survival. Did the authors correlate genes of Cluster C subtype with Classical Score as a negative control for analysis of EMT in pancreatic cancer?

We did not observe a correlation of some genes associated with the Cluster C with the classical score.

However, we must note that these genes do not necessarily can be consider as negative controls in the analysis of EMT in single cell RNA-seq. Our analysis show that the Cluster C is associated with samples with a low proportion of tumor cells, suggesting that the genes associated with this cluster are expressed mainly by the stromal compartment. Indeed, if we analyze the expression of genes associated Cluster C (Figure 2) by scRNA-seq, we observe that they are mainly expressed by stromal cells. *LGALS2* can be found mainly in Acinar cells, *ST6GAL1* in Endothelial and B Cells, *ST6GALNAC2* in Normal ductal cells and *ST6GALNAC6* in Fibroblasts and Stellate cells.

3. Regarding the identification of Fucosylated subtype by plant lectins, both lectins recognize alpha-L-fucose, however, there are some differences seen between the % of positive cells between UEAI and LTA, what can the authors attribute this difference to? Or is it insignificant? Is there a standard deviation (SD) reported?

Despite that both UEAI and LTA are specific alpha-L-fucose, they have preference for different fucosylated structures. UEAI mainly binds to 1,2 fucosylated glycans, while LTA have preference for alpha-L-fucose with a 1,3/4 linkage. These details may explain the differences between the recognition of both lectins observed in cell lines.

4. The glycodendrimer synthesis can benefit from the inclusion of a scheme (in the Supplemental Materials section) and more info about the characterization of these dendrimers. Ref. 33 does not provide enough info either.

We have included a new Supplementary Figure 7 to show the dendrimer synthesis and their recognition by lectin receptors.

REVIEWERS' COMMENTS:

Reviewer #1 (Remarks to the Author):

The authors have made enough changes to the manuscript for publication. Congratulations on well presented piece of work.

Reviewer #3 (Remarks to the Author):

The authors have addressed my comments satisfactorily. I don't have any further questions or suggestions.